

# Spatial variations of microbial communities in abyssal and hadal sediments across the Challenger Deep

Guojie Cui[1,2], Jun Li[1], Zhaoming Gao[1] and Yong Wang[1]

[1] Institute of Deep-Sea Science and Engineering, Chinese Academy of Sciences, Sanya, Hainan, China
[2] University of Chinese Academy of Sciences, Beijing, China

## ABSTRACT

Microbial communities in hadal sediments are least explored in hadal zone (>6,000 m), especially in the Challenger Deep with high pressure (~110 M pa at the bottom). In this study, we investigated the microbial communities in the sediments of the slope and trench-axis bottom of the Challenger Deep in the Mariana Trench. Classification of the reads of the 16S rRNA gene amplicons showed vertical distribution of prokaryotic microbial inhabitants from the surface to up to 60 centimeter below surface floor (cmbsf). The most dominant phyla were Proteobacteria, Chloroflexi, Actinobacteria, Planctomycetes and candidate phyla Patescibacteria and Marinimicrobia. Distinct dominant groups in the microbial communities were observed in trench-axis sediment (water depth >8,600 m), compared to the slopes of the Challenger Deep. A sampling site at the northern slope was enriched with archaea from mesophilic Euryarchaeota Marine Group II (MGII) as a biomarker of specific geochemical setting. Among archaeal community, Thaumarchaeota represented by *Nitrosopumilus* were dominant in the upper layers and diminished drastically in the deeper layers. "*Ca.* Woesearchaeota", however, became the dominant group in the deeper layers. Overall, our study provides a better understanding on the pattern of the microbial communities in the deepest hadal sediments on Earth, and highlights the extraordinary diversity still waiting to be discovered.

## INTRODUCTION

Hadal Trenches at >6,000-m depth are specific ocean ecosystems, and the Challenger Deep isolated from other trenches in the Western Pacific is the deepest place on Earth (*Jamieson et al., 2010*). Subduction plate (seamount tunneling) steepens the forearc, causing the northern slope steeper than the southern slope. The bottom of the Challenger Deep is 11 km long and 1.6 km wide, known as slot-shaped depression. As the slope sediments reach the maximum attainment, sediments slip off the slope. Sediment transport can also be triggered by debris flows and turbidity currents (*Jamieson et al., 2010*). The oligotrophic hadal waters in the Challenger Deep harbored detrital matter degrading microorganisms such as Chloroflexi and candidate phylum Marinimicrobia (SAR406) (*Nunoura et al., 2015*). In contrast, benthic oxygen consumption rate has been

Corresponding author
Yong Wang, wangy@idsse.ac.cn

in situ measured in sediment sites of three hadal trenches (*Glud et al., 2013*; *Wenzhofer et al., 2016*), indicating that the seafloor microbial carbon turnover at the trench-axis was higher than those adjacent abyssal sites probably due to the slot-shaped trench-axis trapped more particulate organic matter than previously estimated (*Glud et al., 2013*; *Nunoura et al., 2015*). Along plate collision, active hydrothermal submarine volcanoes are distributed in the Mariana arc (*Baker et al., 2008*). Previous studies reported a slightly elevated salinity at the water depth above 9,000 m, indicating that the microbial communities could also be shaped by the bottom water mass density (*Nunoura et al., 2015*; *Taira, Yanagimoto & Kitagawa, 2005*). Therefore, with these geochemical, geophysical and geographical parameters, unique microbial communities might have been discovered at the certain slope sites, providing opportunities to develop ecological and evolutionary theories about speciation and community assembly.

It has been speculated that hadal adapted strains of microbes were distinguished from their close relative strains dwelling in shallow marine major due to biological barriers (*Gallo et al., 2015*; *Kato et al., 1998*; *Nunoura et al., 2015*; *Wang et al., 2019b*). Dominance of hadal specific ammonia-oxidizing archaea (AOA) (*Wang et al., 2019b*) provides a hint to the presence of various novel microbial groups in the Challenger Deep, especially at water depth >10,000 m. Some eukaryotic microbial communities in waters from the Challenger Deep were reported recently (*Guo et al., 2018*; *Xu et al., 2018*), suggesting novelty of the eukaryotes in the Mariana Trench. Currently, most studies have focused on the microbial communities in waters rather than those in sediments in the Mariana Trench probably due to difficulties in sediment sampling in this area (*Nunoura et al., 2015*; *Tamburini et al., 2013*; *Tarn et al., 2016*; *Tian et al., 2018*; *Wang et al., 2019a*). At present, the prokaryotic microbial community in a trench-axis bottom site (10,300 m) of the Challenger Deep had been briefly investigated, and revealed that the major phyla were Chloroflexi, Bacteroidetes, Planctomycetes, "*Ca.* Marinimicrobia" (SAR406), Thaumarchaeota and "*Ca.* Woesearchaeota" (*Nunoura et al., 2018*). With the vertical distance of ∼4,000 m in the Challenger Deep, there must be variants in the microbial communities at different sediment sites of the slopes and trench-axis bottom and novel groups could probably be uncovered. Therefore, the spatial distribution pattern of prokaryotes microbial communities in sediments of the Challenger Deep across different hadal depths and sites is still illusive to date.

Aiming to reveal the spatial variations of prokaryotic microbes and find potentially unique groups in the hadal sediments, we investigated five sediment cores from the trench-axis of the Challenger Deep and nine slope sediment cores ranging in depth from 5,480–7,850 m during three cruises in 2016–2017. As the first large-scale study of the microbial communities in sediments of the Challenger Deep, our results deepen our understanding on the formation of unique ecosystem in the hadal zone.

## MATERIALS AND METHODS

### Sample collection

Fourteen sediment samples were successfully collected from the Challenger Deep in three cruises *R/V Dayang 37-II* (DY37II), *Tansuo01* (TS01) and *Tansuo03* (TS03)

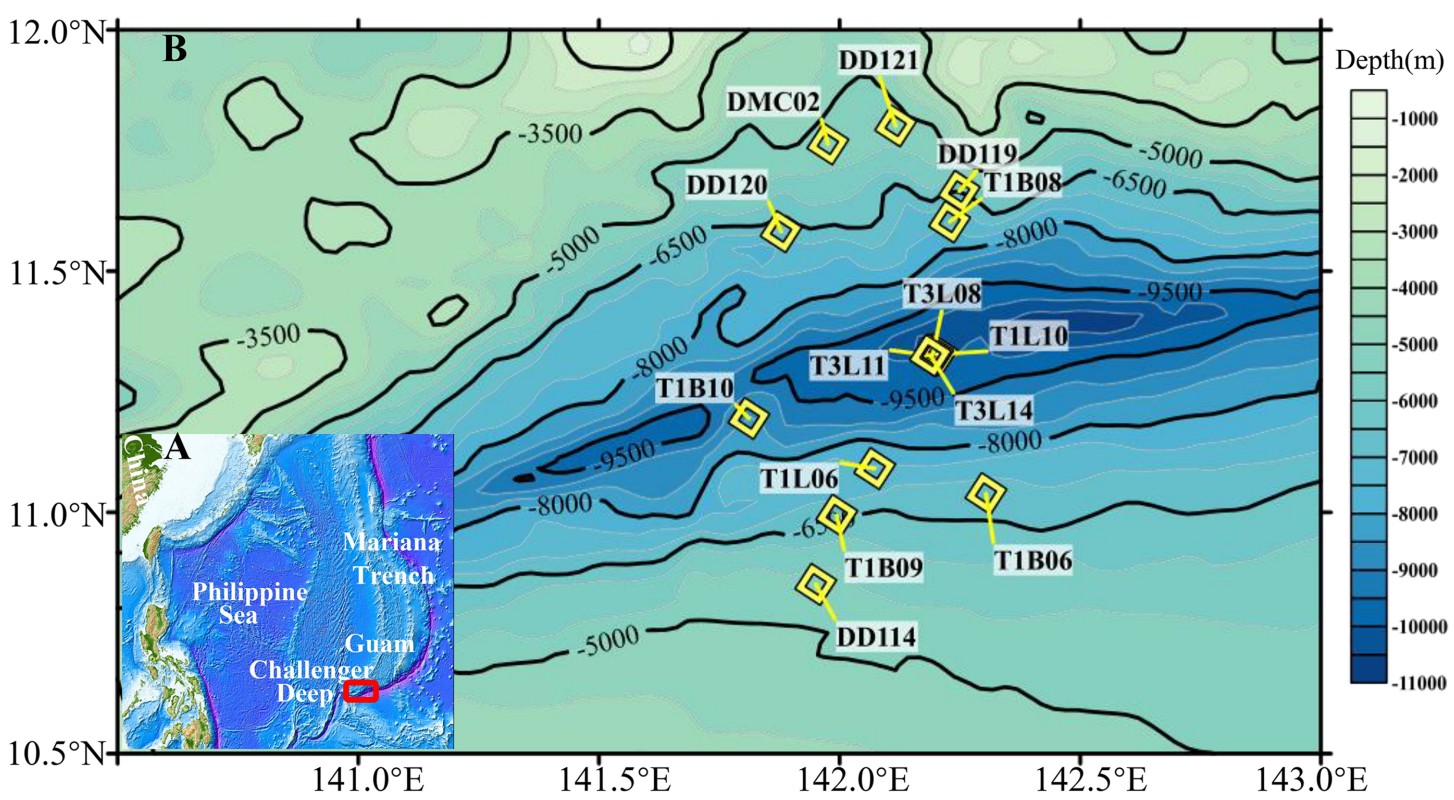

**Figure 1 Sampling sites in the Challenger Deep.** (A) The map was created with Generic Mapping Tools (GMT) (*Wessel & Smith, 1998*). The sampling sites from Challenger Deep were within a red frame in the map. (B) The sediment (yellow diamond) samples were collected during three cruises (R/V TS01, TS03 and DY37II). The sediment samples (water depth >5,000 m) were collected by manned submersible, hadal lander and box-core sampler. Details for the samples were listed in Table S1.

during June–July of 2016, June–August of 2016 and January–March of 2017 (Fig. 1). Our sediment sampling was carried out on the slopes and the trench-axis bottom sites of the Challenger Deep from abyssal to hadal depths by "Jiaolong" manned submersible, a box sampler and a hadal lander (Fig. 1; Table S1). All these sediment cores were sliced into 2 cm subsamples except for three cores (T1L10, T3L08 and T3L11 above 10,000 m) that were sectioned into 3 cm subsamples. The ion concentrations of two sediment cores were measured with an Ion chromatography system (Dionex Corporation, Sunnyvale, CA, USA). All subsamples for community analysis were preserved at −80 °C until DNA extraction. The research has been permitted and secured from the Federated States of Micronesia.

## DNA extraction and quantitative PCR (qPCR) analyses

Genomic DNA was extracted from 2 g of sediment subsamples using the PowerSoil DNA Isolation Kit (MO BIO Laboratories, Inc., Carlsbad, CA, USA). The quality and quantity of the genomic DNA were checked with a NanoDrop spectrophotometer (ND-1000, Nanodrop Technologies, Wilmington, DE, USA) and gel electrophoresis. The abundance of 16S rRNA was quantified as an average of three replicate analyses. The prokaryotic SSU rRNA gene was quantified using the primer Uni516F/Uni806R

using the StepOnePlus™ Real-Time PCR System (Applied Biosystems, Foster City, CA, USA). The primer sequences and qPCR conditions were as described previously (*Nunoura et al., 2018*).

## PCR amplification of the 16S rRNA genes

The 16S rRNA genes were amplified with a pair of universal primers: 341F (5′-CCTAY GGGRBGCASCAG-3′) (*Zakrzewski et al., 2012*) with a tagged six-nucleotide(nt) barcode and reverse primer 802R (5′-TACNVGGGTATCTAATCC-3′) (*Nossa et al., 2010*) that target V3-V4 variable regions. The PCR reaction was prepared according to the PrimerSTAR® HS DNA Polymerase (TaKaRa, Dalian, China) with 2 µl of forward and reverse primers (10 µM) and 2 ng of template DNA. The PCR was performed on a thermal cycler (Bio-Rad, Hercules, CA, USA) in the following thermal cycles: 98 °C for 10 s, 26 cycles of 98 °C for 10 s, 50 °C for 15 s, 72 °C for 30 s and a final extension at 72 °C for 5 min. PCR products were purified using the TaKaRa Agarose Gel DNA Purification Kit (TaKaRa, Dalian, China) and Qubit 2.0 Fluorometer (Invitrogen, Carlsbad, CA, USA) for quantification.

## Analysis of the barcoded amplicons and taxonomic assignment

An equal amount of PCR products from different samples was mixed together and subjected to sequencing on the Illumina Miseq platform ($2 \times 300$ bp) in accordance with the manufacturer's recommendation. The sequencing data were processed and analyzed using QIIME version 1.9.1 (*Caporaso et al., 2010b*). After qualification, the remaining reads were assigned to operational taxonomic units (OTUs) at 97% similarity level by UCLUST (*Edgar, 2010*). Chimeric reads were identified and excluded using ChimeraSlayer (*Haas et al., 2011*) and singleton OTUs with one sequence were removed. For the remaining OTUs, the most abundant read of each OTU was selected as a representative for subsequent taxonomic classification. Taxonomic assignment was conducted using the Ribosomal Database Project (RDP) classifier (version 2.2) (*Wang et al., 2007*) by referring to the SILVA132 database with a confidence level of 80%. The remaining representatives were aligned by PyNAST (*Caporaso et al., 2010a*) and a phylogenetic tree was built using FastTree (*Price, Dehal & Arkin, 2009*). Subsequently, alpha diversity was calculated with three parameters: observed OTUs, Chao1 and Shannon Index. Using percentages of the genera in the microbial communities, Bray–Curtis dissimilarity between the layers was calculated and used for Principal coordinate analysis (PCoA) with the OmicShare tools, a free online platform for data analysis (www.omicshare.com/tools).

## Phylogenetic analysis of 16S rRNA genes

The representative reads of the most abundant OTUs (DatasetS1.fasta) were selected for construction of phylogenetic trees with reference sequences from the Silva and NCBI database. All the sequences were aligned using MAFFT 7.31 (*Katoh & Standley, 2013*) and the maximum-likelihood (ML) phylogenetic trees were constructed using RaxML (*Silvestro & Michalak, 2012*) with GTRGAMMA model. The bootstrap values were calculated based on 1,000 replications. The phylogenetic outputs were visualized and edited in iTOL (*Letunic & Bork, 2016*).

## RESULTS

### Microbial abundance

In the qPCR analysis, the abundance of the entire prokaryotic SSU rRNA gene ranged between $1.5 \times 10^5$ and $5.5 \times 10^8$ copies g$^{-1}$ sediment from sediment surface to 66 cmbsf. The results showed that, with a few exceptions, microbial abundance generally decreased with increasing depth of sediment layers. The SSU rRNA gene copy numbers of the trench-axis at the surface layer (0–2 cmbsf) were observed at least an order of magnitude higher than those of the slopes (Fig. S1). Moreover, the abundance of the entire prokaryotic SSU rRNA at the trench-axis shown in this study was higher than the result reported in a recent work (Nunoura et al., 2018).

### Composition, diversity and species richness of microbial communities

After quality control and filtration of chimeras and singletons, a total of 624,821 reads of 16S rRNA gene amplicons for 95 sediment layers were obtained (Table S2). The rarefaction of the qualified reads resulted in the minimum 1,143 sequences per sample (Table S2). All the curves did not reach the plateau (Fig. S2), indicating that more species might be discovered with more sequencing reads. The qualified reads were grouped into 123,955 OTUs and were then classified into 69 phyla. The 11 most dominant bacterial phyla (>1%) were Proteobacteria, Chloroflexi, Actinobacteria, Planctomycetes, candidate phylum Patescibacteria, candidate phylum Marinimicrobia, Gemmatimonadetes, Bacteroidetes, Firmicutes, Acidobacteria and candidate phylum Zixibacteria (Fig. 2). The most abundant Proteobacteria in hadal sediment samples was Gammaproteobacteria composed of *Pseudoalteromonas* (similarity 100% to NR_152003), *Halomonas* (similarity 100% to NR_043299), *Pseudomonas* (similarity 100% to NR_159318) *and Alteromonas* (similarity 100% to NR_148755). These taxa varied in their percentages spanning 0–63% across all sediments layers. T1B08 was particularly enriched with Gammaproteobacteria highly represented by *Alteromonas*. Planctomycetes and Thaumarchaeota were significantly negatively correlated in their relative abundances (Correlation test with 95 samples; $P < 0.00001$). Two archaeal phyla were abundant in the samples: Thaumarchaeota and Nanoarchaeaeota. *Nitrosopumilus* of Thaumarchaeota was the major group in the Challenger Deep sediment layers, while the latter represented by "*Ca.* Woesearchaeota" was abundantly detected in deep layers of the deepest sediments (>10,000 m). The high abundance of Euryarchaeota archaea was exclusively shown in the layers of the T1B08 sample (Fig. S3).

The observed OTUs, Chao1 and Shannon index were normalized based on the minimal number of reads (Table S2). The sediments from <6,000-m depths were associated with the highest Chao1 and Shannon index such as in DMC02, DD114 and DD121. The diversity indexes for the different layers in the T1B08 were the lowest among the sediments. The communities in the bottom sediments >10,000 m depths were not more diversified than those in the slopes except for the T1B08. Moreover, from the surface to the bottom layers, the biodiversity indexes did not vary notably for all the samples.

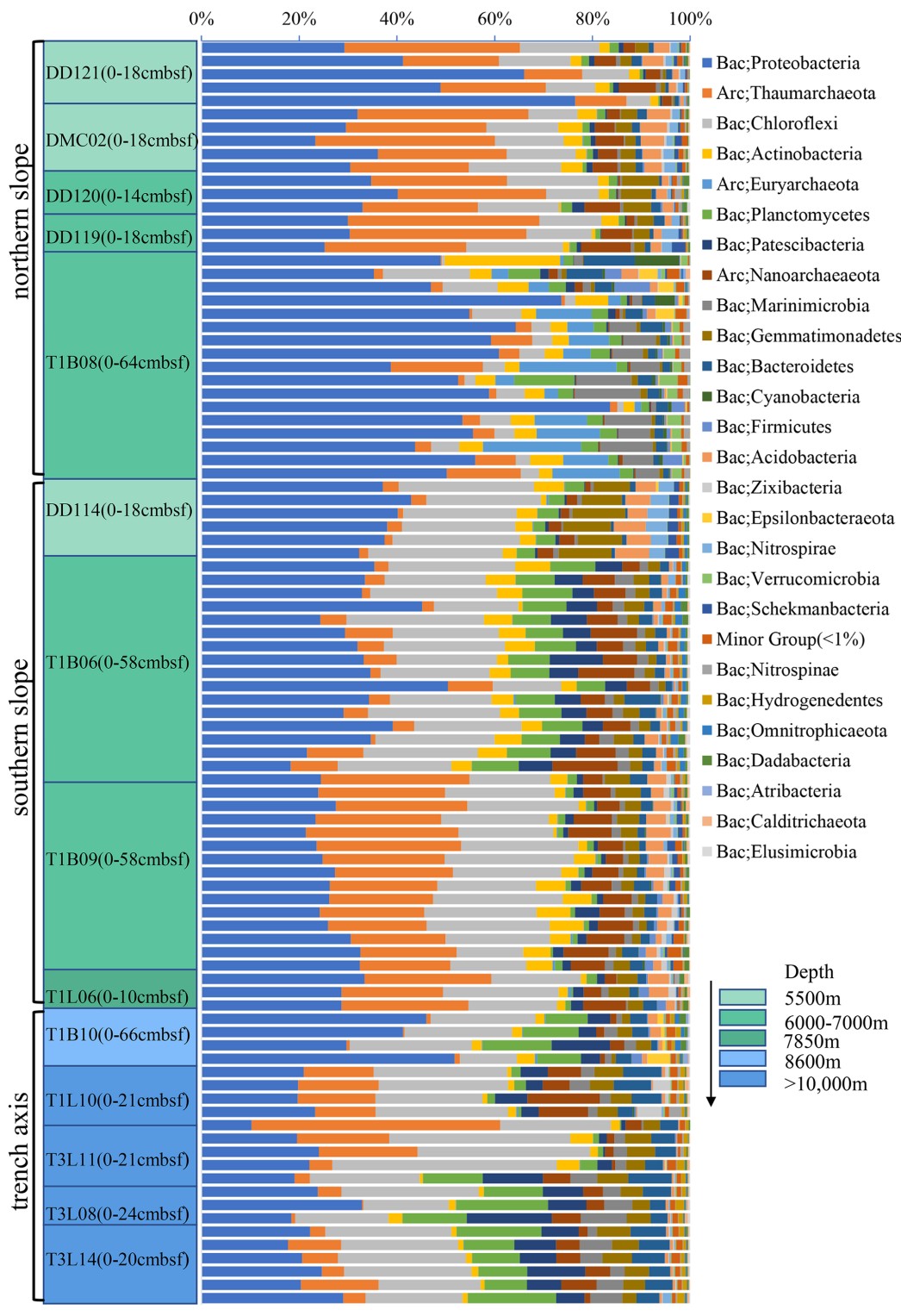

**Figure 2 Relative abundance of prokaryotic phyla in the sediments from the Challenger Deep.** The microbial communities were revealed based on sequencing of 16S rRNA gene amplicons and classifcation at the phylum level using the RDP classifier against the SILVA 132 database. The core lengths were indicated in the brackets behind the sample ID.

## Clustering relationships of microbial communities

A PCoA plot could separate the microbial communities into two groups (the surface 0–2 cmbsf layer samples were shown, simple IDs refer to Table S3), one composing of the slope samples (5,400–7,800 m) and another containing only trench-axis bottom samples (8,600–11,000 m). In the former group, the samples from the northern slope could be further split from those from the southern slope. The T1B08 sample was far from the two groups in the plot (Fig. 3A), reflecting its distinct composition of microbial community. Principally, the microbial communities were clustered based on the geographic locations and depths. The combination of PCO1 and PCO2 explained 53% of the differences among the communities (Fig. 3A). Hierarchical clustering of the samples using the percentages of genera in the corresponding communities also demonstrated that the microbial communities in trench-axis sediments differed from those in the slopes (Fig. 3B). The similarity of the microbial communities was determined by sampling sites, rather than depths to the surface layers, since the communities from the same cores tend to be grouped together (Fig. S4).

## Phylogenetic analysis of 16S rRNA genes

The most abundant OTUs (DatasetS1.fasta) were used to determine their phylogenetic positions. The representative reads for a total of 35 OTUs were selected for the construction phylogenetic tree with their closest relatives in the NCBI. The most abundant one was within the same group with a neighbor of *Nitrosopumilus* (Fig. 4A). The *Nitrosopumilus* in the Challenger Deep sediments resembled those inhabiting other abyssal and hadal waters and sediments from the Ogasawara Trench, the Japan Trench and even the Puerto Rico Trench, with respect to their short phylogenetic distance. This has been observed and discussed in a recent work on the high homogeneity of *Nitrosopumilus* AOA in hadal zones (*Wang et al., 2019b*). The phylogenetic position of OTU100704 particularly enriched in T1B08 showed high affinity (100% identity) to mesophilic archaea of Marine Group II (MGII) (estimated optimum growth temperature: $T_{opt.}$= 40.76 (*Kimura et al., 2013*)) isolated from the Juan de Fuca hydrothermal flume (*Anderson et al., 2013*). The five OTUs affiliated with Nanoarchaeaeota are *Ca.* Woesearchaeota widely distributed in the Challenger Deep sediments from 5,481 to 10,953 m. Similar sequences were also obtained from saline water. The five OTUs in our results can split into two groups, and thus there would be more potentially novel species in this phylum.

Actinobacteria, Cyanobacteria, Firmicutes and Chloroflexi as members of superphylum Terrabacteria are almost ubiquitous in the sediments. The six OTUs of Chloroflexi in ML phylogenetic tree were split into four major groups (Fig. 4B), in which OTU124775 seems to represent a novel taxon. Interestingly, OTU344095 was approximate to *Dehalococcoides* spp., a known genus of strictly anaerobic bacteria with capacity of gaining energy from the reduction of chlorinated compounds (*Tas et al., 2010*). Furthermore, "*Ca.* Actinomarinales", described as a new group of photoheterotrophic Actinobacteria with ubiquitous presence in the pelagic layer of the oligotrophic ocean (*Ghai et al., 2013*), was also found in sediment layers especially in surface layer of T1B08. Furthermore, a proportion of 16S rRNA OTUs in the surface layer of T1B08 belonged to the

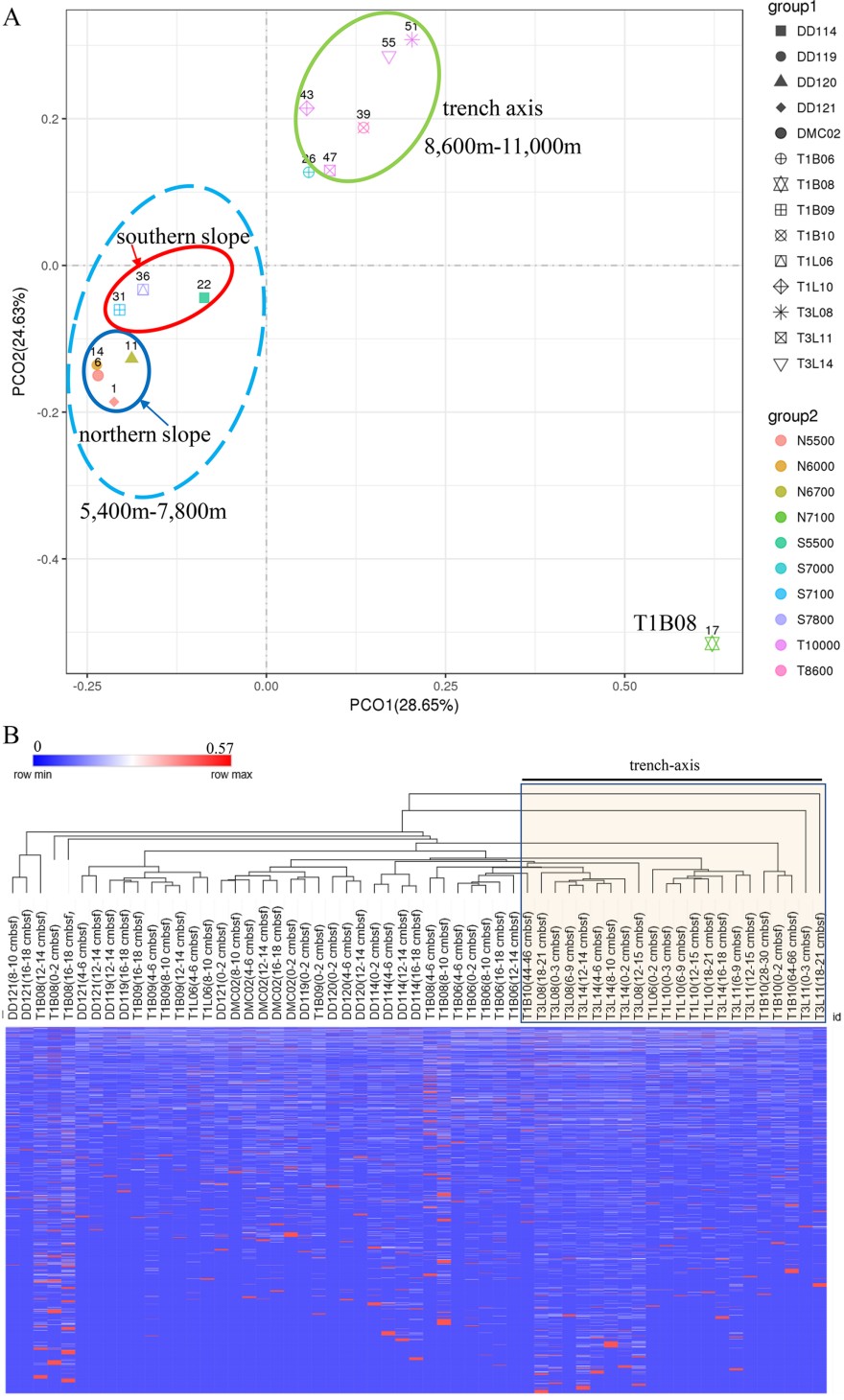

**Figure 3 Principal coordinates analysis (PCoA) and hierarchical clustering of the samples.** (A) PCoA of the microbial communities. The percentages of genera in the communities were used for calculation of Bray–Curtis dissimilarities and then a PCoA plot. Color code indicates the different water depths of the samples while different symbols distinguish the samples from each other (simple IDs refer to Table S3). (B) Hierarchical clustering of the samples was based on the Bray–Curtis dissimilarities. The samples from the trench-axis of the Challenger Deep were framed and labeled.

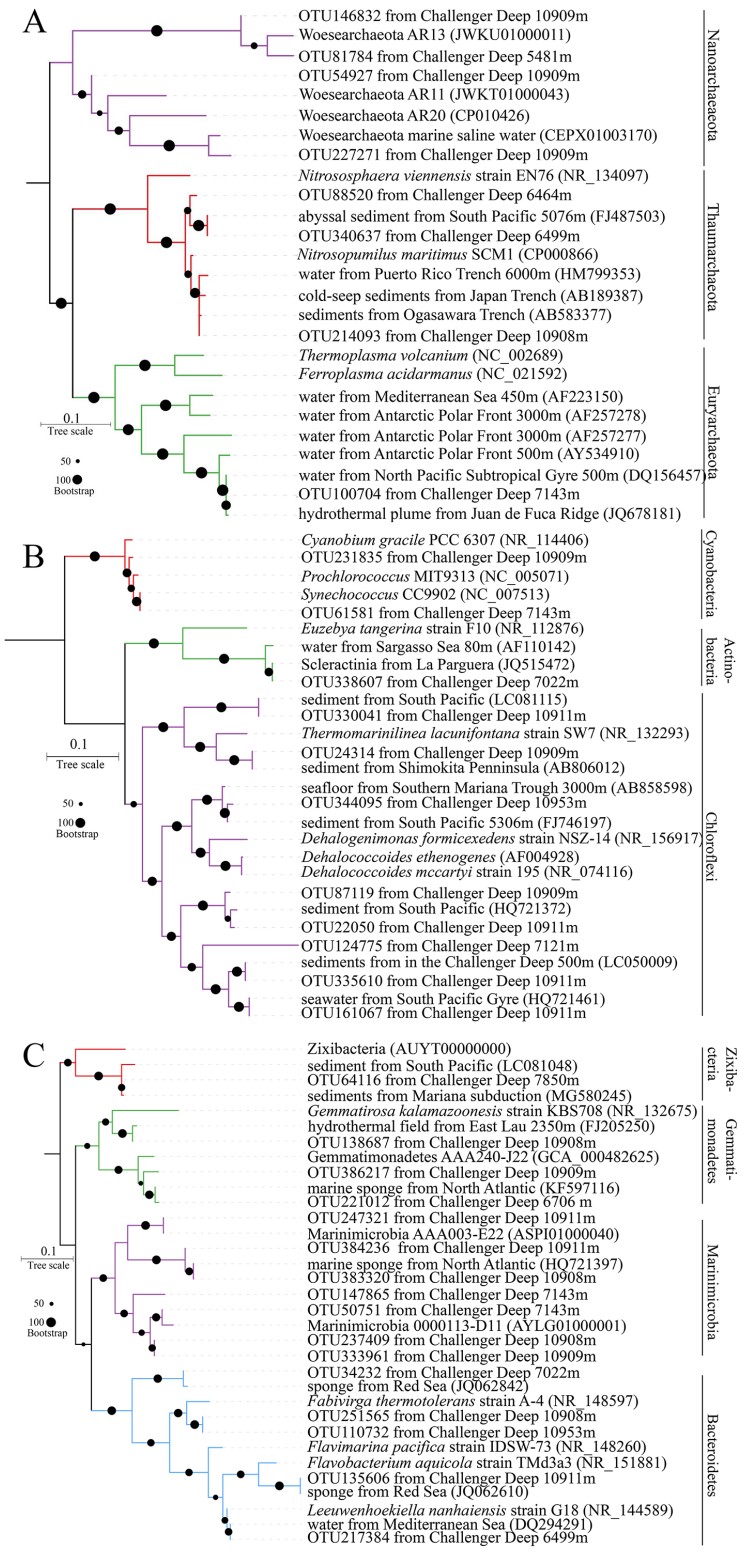

**Figure 4 Phylogenetic trees for representative reads of 16S rRNA OTUs.** The bootstrap supports were based on 1,000 replicates. The OTUs were most abundant in the microbial communities. Their representative reads were collected to construct an ML tree ((A) for archaea; (B) and (C) for Terrabacteria and FCB superphylum, respectively).               

picophytoplankton *Synechococcus* (3%) and *Prochlorococcus* (7%) in Cyanobacteria phylum (Fig. 4B). The two Cyanobacteria genera and "*Ca.* Actinomarinales" originally present in the euphotic layer were rarely detected in the other sediment samples. The abundance of these euphotic bacteria in T1B08 surface layer was probably buried relics from depositional material (*Kirkpatrick, Walsh & D'Hondt, 2016*) or might be derived from contamination during the operation of the box-core sampler.

FCB (Fibrobacteres, Chlorobi, Bacteroidetes) is another superphylum mainly containing Chlorobi, candidate phylum Marinimicrobia and Bacteroidetes (*Bertagnolli et al., 2017*; *Getz et al., 2018*; *Hug et al., 2016*). Nowadays, more new taxa within the FCB superphylum were recovered from sediment metagenomes (*Castelle et al., 2013*; *Parks et al., 2017*; *Rinke et al., 2013*), indicating their prevalence in various environments on Earth. In this study, OTU64116 was grouped with references of a newly defined phylum Zixibacteria (*Castelle et al., 2013*). Some of the Marinimicrobia OTUs were grouped with the sequences from previous studies, while the others represented novel groups that might have evolved in the hadal sediments (Fig. 4C).

## DISCUSSION

In this study, we revealed spatial variations of the microbial communities in the sediments from the slopes and the trench-axis bottom of the Challenger Deep. Here, we also figure out that hadal communities within the trench-axis are distinct from the hadal slopes by their major taxa and the potentially unidentified OTUs, such as some groups in Chloroflexi and candidate phylum Marinimicrobia. It is believed that the organic matter accumulation rate in sub-sea floor sediment attenuated with the increase of water depth, however, hadal trenches do not comply with this rule despite oligotrophic water column above (*Luo et al., 2017*). Recently, benthic oxygen consumption rate has been successfully in situ measured (*Glud et al., 2013*; *Luo et al., 2018*; *Wenzhofer et al., 2016*), indicating that the seafloor oxygen consumption rates and the microbial carbon turnover at the trench-axis were higher than those adjacent abyssal sites. Furthermore, the funneling effect and erratic downslope sediment transport within the hadal trenches lead to fresher and more labile organic matter concentrated at the trench-axis bottom based on sedimentation rate that was higher than global average, together with total organic carbon contents increasing with water depth (*Glud et al., 2013*; *Turnewitsch et al., 2014*). Therefore, hadal trench-axis area is a natural vector to capture organic matter, which ultimately contributes to the different microbial communities compared to those in the slopes.

Recently, a bottom sediment core at depth of 10,300 m was investigated and geochemical data were provided by *Nunoura et al. (2018)*. Our measurements of nitrate and sulfate in two cores (Table S4) from the trench-axis bottom were consistent with their results, but the validity of the data was questioned majorly due to decreased hydrostatic pressure. Therefore, the bottleneck of the geochemical work at such depths is to make sensors for the in situ work (*Glud et al., 2013*; *Luo et al., 2018*). To date, only trench-axis bottom oxygen consumption rate was in situ measurement at that hadal depth. In this study, we did not conduct chemical and nutrient analyses on all the sediment samples, but the

dominant species resembling currently known ones could provide some hints to the local environments. Strikingly, the phylogenetic tree exhibited that most of the dominant species were grouped with known species, suggesting that these species were widespread in the trench sediments, and even in trench waters. The SAR202 bacteria as the most abundant group in Chloroflexi were degraders of various detrital materials in the deep-sea zones (*Landry et al., 2017*). We also revealed that potential novel group of Chloroflexi dwelling in the Challenger Deep. The genomics analysis of the Chloroflexi in hadal waters coincides with the previous studies on its recycling capacity (*Landry et al., 2017*; *Mehrshad et al., 2017*). The groups from candidate phylum Marinimicrobia were also abundant with their ability to recycle peptides and nitrogen compounds (*Hawley et al., 2017*). Furthermore, trenches are the habitat to specific microbial communities with specificities for pressure adaption such as the known piezophilic genera *Colwellia*, *Shewanella*, *Moritella* and *Psychromonas* (*Bartlett, 1992*). In this study, these major piezophilic groups of Gammaproteobacteria have already been found in our samples as well as in other hadal surface sediments (*DeLong, Franks & Yayanos, 1997*; *Kato et al., 1998*; *Nogi et al., 2004*; *Xu et al., 2003*; *Zhang et al., 2018*), indicating its important ecological role in hadal sediments for detrital carbon recycling. The dominant species in Planctomycetes was "*Ca.* Scalindua", which is capable to reduce nitrite with ammonia in anoxic condition (*Schmid et al., 2003*; *Woebken et al., 2008*). *Nitrosopumilus* as an alternative ammonia oxidizer is a typically aerobic since the ammonia oxidization process in the archaea requires oxygen. A recent study showed a quick descent of $O_2$ concentration in a surface sediment sample (10,817 m) (*Glud et al., 2013*). This explains the negative correlation in relative abundance between Planctomycetes and Thaumarchaeota in our samples.

In this study, a large number of the sediment OTUs were categorized as Nanoarchaeaeota (Woesearchaeota) phylum. Previous researchers revealed that AR20 was one of the most widely distributed "*Ca.* Woesearchaeota" (*Castelle et al., 2015*). With the first complete genome, AR20 genome was only 0.8 Mb with incomplete core pathways, suggesting a potential symbiotic or parasitic lifestyle (*Castelle et al., 2015*). Additionally, the "*Ca.* Woesearchaeota" found in the hadal sediments also preferred the deeper layers (*Nunoura et al., 2018*), suggesting that oxygen was the determining factor that affects its distribution. Moreover, Nanoarchaeaeota (Woesearchaeota) could split into at least two groups, suggesting that there should be more unidentified groups in the phylum and worthwhile to be further explored in the hadal zone. In some layers of the T1B08, MGII was one of the dominant groups. As active hydrothermal submarine volcanoes were widely distributed in the Mariana arc (*Baker et al., 2008*), it should not be surprising to find mesophilic archaea of MGII in T1B08 as a result of hypothetical hydrothermal process occurring in the sediments. Hence, there are more microenvironments located on the northern slope for the formation of unique but sporadic microbial communities.

For the potential unidentified groups detected in this study, their roles remain unknown. Perhaps, future omics work may provide their taxonomic positions and metabolic features. In addition, the hadal trenches have their own characteristics.

For example, the Japan Trench situated in the Kuroshio Current province is relatively eutrophic (*Jamieson et al., 2009*; *Nakatsuka et al., 1997*) and the South Sandwich Trench is the only sub-zero hadal zone (*Vanhove, Vermeeren & Vanreusel, 2004*). The hypothesis of the microbial inhabitants as the markers to the sedimentary process and in situ environments is perhaps not applicable to the other trenches.

The rarefaction curves and estimates of Chao1 and Shannon were indicative of insufficient sequencing depth for recovery of the microbial communities in our sediment samples. To date, sequencing of 16S rRNA gene amplicons has been widely applied to study microbial communities in various environmental conditions. Selection of proper primers is one of the important factors for detection of a microbial community with high fidelity. In this study, the 341F and 802R primers for V3-V4 region were evaluated to recover 86.4% of Bacteria and 78.7% of Archaea in the Silva database although they have been considered as the best universal primer pair (*Lu et al., 2015*). However, it is worth noting that the surveys based on amplification of 16S rRNA genes are always imperfect. Sampling time and variable geographic differences may also affect the detection of certain taxa.

## CONCLUSIONS

Our analysis first provides spatial distribution pattern of prokaryotes microbial communities in sediment of the Challenger Deep. We also figure out the communities within the trench-axis bottom are distinct from the hadal slopes, and special microbial communities were presented at certain sites. Furthermore, some novel groups were identified based on phylogenetic trees. Overall, our results deepen our understanding on microbial ecosystem in the hadal zone.

## ACKNOWLEDGEMENTS

We are grateful to the team members aboard the *R/V Tansuo01* and *Tansuo03* for their safe navigation and their invaluable efforts in the sampling cruises. Great thanks are given to J. Chen, Y. Z. Xin and D.S. Cai for their help in sampling.

### Funding

This study was supported by the National Key Research and Development Program of China (No. 2016YFC0302500), the National Science Foundation of China (Nos. 41476104 and 31460001), the Youth Innovation Promotion Association Program of Chinese Academy of Sciences (Y710071001) and the Strategic Priority Research Program B of Chinese Academy of Sciences (Nos. XDB06010201 and XDB06040101). The funders had no role in study design, data collection and analysis, decision to publish, or preparation of the manuscript.

### Grant Disclosures

The following grant information was disclosed by the authors:
National Key Research and Development Program of China: 2016YFC0302500.

National Science Foundation of China: 41476104 and 31460001.
Youth Innovation Promotion Association Program of Chinese Academy of Sciences: Y710071001.
Strategic Priority Research Program B of Chinese Academy of Sciences: XDB06010201 and XDB06040101.

## Competing Interests

The authors declare that they have no competing interests.

## Author Contributions

- Guojie Cui conceived and designed the experiments, performed the experiments, analyzed the data, prepared figures and/or tables, approved the final draft.
- Jun Li controlled the lander for collecting sediment samples.
- Zhaoming Gao contributed reagents/materials/analysis tools.
- Yong Wang authored or reviewed drafts of the paper.

## Field Study Permissions

The following information was supplied relating to field study approvals (i.e., approving body and any reference numbers):

Research and training permits was provided by the Federated States of Micronesia.

## Data Availability

The OTUs sequences used in this study are available as a Supplemental File. Our raw data contained other samples not presented in this study.

## Supplemental Information

Supplemental information for this article can be found online at http://dx.doi.org/10.7717/peerj.6961#supplemental-information.

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
