# Peer review of "Spatial variations of microbial communities in abyssal and hadal sediments across the Challenger Deep"

_PeerJ, doi:10.7717/peerj.6961_

## Round 0.1 · original submission · Major Revisions

Dear Dr. Cui and colleagues:

Thank you for submitting your manuscript to PeerJ. I have now received two independent reviews of your work, and as you will see, the reviewers raised some concerns about the manuscript. Despite this, these reviewers are optimistic about your work and the potential impact it will have on research communities studying microbial diversity in deep trench ocean environments. Thus, I encourage you to revise your manuscript accordingly, taking into account all of the concerns raised by both reviewers.

In your revision, please be transparent on the methods for comparative analyses, and also improve the overall clarity of your methods. One reviewer would like to see the inclusion of “geochemical data…quantitative microbiology data (such as cell abundance and quantitative PCR for 16S rRNA gene)”. I agree that this data would greatly improve your analyses and interpretations. Also, both reviewers identified many missing references, and also areas where the clarity of the manuscript can be improved. Please address all of these issues.

Please note: while most of the concerns of the reviewers are relatively minor, this is a major revision to ensure that the original reviewers have a chance to evaluate your responses to their concerns.

I look forward to seeing your revision, and thanks again for submitting your work to PeerJ.

Good luck with your revision,

-joe

Reviewer 1 ·

Basic reporting

The manuscript presents some interesting, relevant and very timely data. However, the manuscript tends to be very descriptive and - in my opinion – it lacks a clear well defined research aim. What is the hypothesis that is going to be tested, evaluated and discussed? Without a clear focus, the presentation and discussion becomes very descriptive and also speculative – and in my opinion the “conclusions” are rather weak and not substantiated by the presented data.
The wording is often impressive and the authors should streamline the text being more explicit and precise when they discuss and evaluate their data.
In conclusion I recommend acceptance after major revision – most importantly the revised manuscript should provide a clear research focus with a well formulated aim (hypothesis) which could provide the manuscript some direction.
Below please find a list of concrete suggestions.
Abstract
Line 39 “dramatic” is in my opinion an exaggeration – I suggest eliminating “Dramatic” from the sentence.

Line 46; What do you mean by “novel” – please be specific. As written, I am not sure that this claim is substantiated by the presented data. Also I find the terminology “potential endemic unidentified Clades” very speculative. Please tone down these claims – and simply just write what you have found with using word like “potential endemic”. Such statements would require additional physiological, biochemical and biogeochemical measurements.

Inroduction
Line 67-68, Improve wording, the grammar is incorrect and the wording unfortunate. Also the statement “with the effect of water current…” is very unspecific - what do you mean?

Line 71-75…; wrong tense – this makes it confusing. Begin with “Previous studies have shown that…. “. Also the structure is un-logic. The section begin with pelagic microbiology, then cold seeps, then landslides, then back to microbiology at the Northern slope and out of nowhere enhanced salinity at 9000m. Try to streamline and be explicit about what you really want to say when leading up to your study.

Line 87; Please define “ecotype”

Line 88; A recent work… does not really complies with referencing a 10 year old study.

Line 93-97…; reads very inconsistent- In fact the whole paragraph reads like a list of information without really synthesizing consensus or identifying the actual challenges to be tackled by this study and our community in general. What is the hypothesis or the aim of this study?

Line 109-116..; this reads like a conclusion, - though very indirect and unspecific- e.g. How should a funneling effect facilitate distinct microbial communities at depth???? This is unclear. Be specific when you explain. The last sentence is very unclear – in what way does this study promote……???? This paragraph should rather present the specific aim of your investigation.

Materials and Methods
Line 124-125; the core sectioning procedure is not very please include the actual sediment horizons for the specific samples in Figure 2. That would help the reader to directly relate observations to the sediment sampling depth.

Results
Line 172-173; how many more “species” (relative speaking) are you missing?

Line 180; Suppose you mean “specific taxa” – not sure this is very dramatic given that different diagenetic pathways operates in the different sediment layers. If you use a word like “dramatic” you must relate it to something … compared to abyssal reference sites or???

Line 189-191; So why would a facultative anaerob be facilitated by “a oxygen flux that was minimized to the best for…”. This is unclear please be specific on what you mean.

Line 198; what do you mean? This seams opposite to the high O2 consumption – please explain and be specific. It becomes very difficult to begin comparing sediments from different sampling sites without any indications on the ongoing diagenetic activity. So does the variations simply reflect that different processes are ongoing in the respective sediment layers across the depth transects given different carbon courses – and remain unrelated to depth (pressure) as such. What is the aim of this study?

Line 204; do you mean “Nutrition” rather than “nutrients”? Otherwise it is unclear to me.

Line 209; How does the data reflect “its uniqueness”? I can accept it is “separate” or “different”, but how do you now it is “unique”.

Line 229-230; What do you mean by”…potential novel species in this phylum except for W…”?

Line 239…249; So do you consider all the phototropic bacteria to be an artefact of recovery-contamination ? Or are they simply buried relicts from depositional material. This is unclear to me? What do you mean please be explicit and clear in your wording.

Line 262-263; Be careful using words like “Unique” or “endemic” – you do not know that!

Line 262-264 How do you know? Could the extracted DNA be E-DNA ?

Line261-283; In this paragraph you argue for that benthic trench communities are “survivors from the surface” to “endemic communities”.. it is not really clear for me what you mean and what are the specific arguments for the respective “conclusions” – please be more specific. You have absolutely no clue if these play a geochemical important role?

Line 329; So do you argues for methanogenic activity in the deep trench? Or is this just a result of deposition from above (which is a more likely explanation)? Please be clear?

A conclusive section is not needed – this is a very short paper – and punch lines should be clear in the Discussion.

Experimental design

see above

Validity of the findings

see above

Additional comments

see above

Reviewer 2 ·

Basic reporting

L44-46: What do “clades” mean? In my interpretation of clades mostly higher taxonomic clades I did not find any “endemic clades” thorough the manuscript.

L67-68: Please provide appropriate references for the sentence. If not, it should be deleted or the content should be moved to discussion section.

L75-78: Please provide appropriate references for the sentences. If not, they are likely based on speculation of the authors and are not appropriate for the introduction section.

L84: Recently more papers have been published, e.g. Jing et al. 2018; Guo et al. 2018; Xu et al. 2018; Zhang et al. 2018.

L85: Nishizawa et al. 2013 is not appropriate.

L88-91: The information is not consistent with the reference “Lauro and Bartlett”. The sentence is likely incomplete.

L101: What does preliminary mean? This manuscript lacks both geochemistry and quantitative microbiology data sets and is likely preliminary parts of huge project although the authors analyzed numbers of samples.

L113: Appropriate references should be provided for “due to the rapid accumulation of POC under funneling effect.” If not, the sentence is not appropriate in this section.

L119-: I think all the samples were retrieved from the EEZ of Micronesia. Permission of the sampling should be stated here or acknowledgement section.

L131: Appropriate references should be provided for the primer set used in this study.

L174: For the candidatus phyla, they should be presented as “Ca. XXX” through the manuscript. In addition, previous division names are helpful for readers.

L178: Halomonas should be presented by italic.

L190: “suggesting” instead of “indicating”. The manuscript lacks geochemical data set. Thus, “indicating” is not appropriate.

L203-207: Appropriate references should be provided because due to the absence of geochemical data set, the environmental data are probably not enough to conduct complete statistics.

L222-224: The number of OTUs given in Fig 4 is not sufficient to discuss about the connectivity among the trench phylotypes.

L224-226: The result is interesting, but the information is not sufficient. How about the similarity between the OTU to Methanomassiliicoccus? Only the low-esolution phylogenetic tree in Fig 4 is not enough.

L238-244: Appropriate references should be provided.

L247: Identification of 16S rRNA genes from chloroplast has been identified in subseafloor sediments and discussed (e.g. Kirkpatrick et al. 2016). It is not so surprising. Considering the difference in biomass between the surface water and sediments, the abundance of such sequences is probably too high to speculate as contaminants although the authors did not present quantitative data set.

L251: Appropriate references should be provided because the grouping has not been authorized.

L254-255: Appropriate references should be provided.
L263-266: Appropriate references should be provided.

L268-272: Luo et al. 2018 should also been cited. In addition, there are also reports without in situ oxygen concentration measurements that obtain similar trend.

L286: Please explain why did the data indicate elevated “primary” production.

L294: Appropriate references should be provided.

L304-306: Appropriate references should be provided.

L310: Appropriate references should be provided. Spore deposition or dormant cells in the trench bottom has also been suggested by the isolation of thermophilic Thermaerobacter and Geobacillus (e.g. Takai et al. 1999; Takami et al. 2004). On the other hand, Bacillus is also recognized as a typical contaminants in lab analyses.

L317: Nunoura et al. 2016 is not appropriate reference. The authors should cite oceanography papers.

L326: Because the manuscript completely lack geochemical data set, appropriate references are necessary in discussion between microbial components and geochemical parameters.

Experimental design

The manuscript lacks geochemical data set as the authors mentioned while the authors analyzed numbers of 16S rRNA gene diversity. In addition, quantitative microbiology data such as cell abundance and quantitative PCR for 16S rRNA gene. These are usually requested for complete environmental microbiology papers. In the previous papers enumerating microbial activity between trench bottom and adjacent abyssal plain sediments, microbial biomass was reported while microbial diversity analyses were absent. To compare the results in this paper and previous works, I hope to add quantitative data set in this paper.

Validity of the findings

No comment

---

## Round 0.2 · Minor Revisions

Dear Dr. Cui and colleagues:

Thanks for resubmitting your manuscript to PeerJ. I have now received one review of your work, and as you will see, the reviewer still raised some concerns about the manuscript. Despite this, the reviewer is optimistic about your work and believes the revision is much improved. Thus, I encourage you to revise your manuscript accordingly, taking into account all of the concerns raised by the reviewer.

In your revision, please consider the reviewer’s concern about microbial cell counts.

I look forward to seeing your revision, and thanks again for submitting your work to PeerJ.

Good luck with your revision,

-joe

Reviewer 2 ·

Basic reporting

Through the manuscript: Candidatus phylum can be presented as “Ca. XXX” through the manuscript.

L71: “In contrast” instead of “However”.

L93: Pathom-Lee et al. 2006 is not likely appropriate reference here.

L95: What does “known phyla” mean?

L218: There is no evidence that TIB08 MGII phylotype is derived from thermophile in the reference. Sometimes non-thermphilic microbes are detected from samples taken from high temperature environments. Kimura et al. 2013 in Env. Microb. Rep. and other reports are helpful to estimate the growth temperature from 16S rRNA gene sequence.

L270-: It is not appropriate about the impacts of hydrate on geochemical analyses for samples for the ambient deep-sea sediment samples. The sentences should be deleted.

L293: “under” instead of “in”

L293: The occurrence of the anammox can be a signature of the absence or very low concentrations of oxygen. However, there are many reports that the presence of thaumarchaeal 16S rRNA gene sequence from the depths that oxygen concentrations are not detectable in marine sediments. It is not clear whether they are relics or active, it cannot be a signature of oxygen concentration.

L299: Very old paper presenting higher abundance of Bacillus is not reliable, and probably a signature of lab contaminants.

L309: The finding of “Ca. Woesearchaeota” (DHVE6) in deeper layers in deep-sea sediments has been reported before even in the Mariana Trench sediments. Appropriate paper should be cited.

L315: “thermophilic archaea of MGII” see above.

L320: What are the novel groups found in this study?

L460: The authors list is not appropriate.

Table S4: Please provide analytical method.

Experimental design

I understand the difficulty to provide pore-water geochemistry, but I cannot understand microbial cell counts are not presented in this study while the authors have obtained the data set. Only cell count (or qPCR) would contribute to connect this study and previous and future studies for trench benthic microbial ecosystems except for the geological setting. The data set is essential.

Validity of the findings

no comment

Additional comments

The manuscript has been extensively updated, while issues listed above still remain.

---

## Round 0.3 · accepted · Accept

Dear Dr. Cui and colleagues:

Thanks for revising your manuscript based on the minor concerns raised by the reviewers. I now believe that your manuscript is suitable for publication. Congratulations! I look forward to seeing this work in print, and I anticipate it being an important resource for research communities studying microbial diversity in deep trench ocean environments. Thanks again for choosing PeerJ to publish such important work.

Best,

-joe

#